PyMC: a modern, and comprehensive probabilistic programming framework in Python

http://orcid.org/0000-0002-1847-9481 Abril-Pla Oriol 1
http://orcid.org/0000-0002-2383-1478 Andreani Virgile 2 3
Carroll Colin 4
Dong Larry 5 6
Fonnesbeck Christopher J. 7 fonnesbeck@gmail.com
Kochurov Maxim 8
Kumar Ravin 9
http://orcid.org/0000-0001-6029-7510 Lao Junpeng 10
Luhmann Christian C. 11 12
http://orcid.org/0000-0001-7419-8978 Martin Osvaldo A. 13
Osthege Michael 14
Vieira Ricardo 8
Wiecki Thomas 8 thomas.wiecki@pymc-labs.com
http://orcid.org/0000-0002-8654-2235 Zinkov Robert 15
1 ArviZ-Devs , Barcelona , Spain
2 Biomedical Engineering Department, Boston University , Boston , United States of America
3 Biological Design Center, Boston University , Boston , United States of America
4 Google , Cambridge, MA , United States of America
5 Dalla Lana School of Public Health, University of Toronto , Toronto , Canada
6 Child Health Evaluative Sciences, The Hospital for Sick Children , Toronto , Canada
7 Baseball Operations Research and Development, Philadelphia Phillies , Philadelphia , United States of America
8 PyMC Labs , Berlin , Germany
9 Google , Mountain View, CA , United States of America
10 Google , Zürich , Switzerland
11 Department of Psychology, Stony Brook University , Stony Brook , United States of America
12 Institute for Advanced Computational Science, Stony Brook University , Stony Brook NY , United States of America
13 IMASL-CONICET, Universidad Nacional de San Luis , San Luis , Argentina
14 Forschungszentrum Jülich GmbH , Jülich , Germany
15 Oxford University , Oxford , United Kingdom
Aleem Muhammad
Electronic publication date: 2023 Sep 1
Publication date: 2023
Volume: 9
Electronic Location ID: e1516
Received 2023 Jun 2; Accepted 2023 Jul 13
Copyright: © 2023 Abril-Pla et al.
Copyright year: 2023
Copyright holder: Abril-Pla et al.
License: This is an open access article distributed under the terms of the Creative Commons Attribution License, which permits unrestricted use, distribution, reproduction and adaptation in any medium and for any purpose provided that it is properly attributed. For attribution, the original author(s), title, publication source (PeerJ Computer Science) and either DOI or URL of the article must be cited.
License URL: https://creativecommons.org/licenses/by/4.0/

Keywords: Bayesian statistics, Probabilistic programming, Python, Markov chain Monte Carlo, Statistical modeling

Funding: NumFOCUS PyMC Labs National Agency of Scientific and Technological Promotion ANPCyT PICT-02212 National Scientific and Technical Research Council CONICET PIP-0087 NumFOCUS, a nonprofit 501(c)(3) public charity, provides operational and financial support to PyMC. PyMC Labs, a Bayesian consulting company, provides funding for the the development of PyMC. This research was supported by National Agency of Scientific and Technological promotion ANPCyT, Grant PICT-02212 (O.A.M.) And National Scientific and Technical Research Council CONICET, Grant PIP-0087 (O.A.M). The funders had no role in study design, data collection and analysis, decision to publish, or preparation of the manuscript.

==============================
PyMC is a probabilistic programming library for Python that provides tools for constructing and fitting Bayesian models. It offers an intuitive, readable syntax that is close to the natural syntax statisticians use to describe models. PyMC leverages the symbolic computation library PyTensor, allowing it to be compiled into a variety of computational backends, such as C, JAX, and Numba, which in turn offer access to different computational architectures including CPU, GPU, and TPU. Being a general modeling framework, PyMC supports a variety of models including generalized hierarchical linear regression and classification, time series, ordinary differential equations (ODEs), and non-parametric models such as Gaussian processes (GPs). We demonstrate PyMC’s versatility and ease of use with examples spanning a range of common statistical models. Additionally, we discuss the positive role of PyMC in the development of the open-source ecosystem for probabilistic programming.

Introduction

Probabilistic programming languages (PPLs) are general-purpose programming languages or libraries with built-in tools for Bayesian model specification and inference, allowing practitioners to focus on the creation of models rather than on computational details (Rainforth, 2017). PPLs have dramatically changed Bayesian modeling, enabling practitioners to perform analyses of increasing complexity while simultaneously lowering barriers to entry. Additionally, PPLs facilitate an iterative modeling process that is now more relevant than ever (Gelman et al., 2020; Martin, Kumar & Lao, 2021).

Since the early 2000s, the PyMC project has provided scientists with an open-source PPL in Python (Salvatier, Wiecki & Fonnesbeck, 2016). The first release of PyMC version 2.0 in January 2009 introduced a general implementation of the Metropolis-Hastings sampler (Metropolis et al., 1953; Hastings, 1970), accelerated by probability distributions implemented in Fortran. In 2016 a major rewrite of PyMC introduced gradient-based sampling algorithms, notably Hamiltonian Monte Carlo (HMC) and the No-U-Turn-Sampler (NUTS) algorithm (Hoffman & Gelman, 2014) that dramatically improved the speed and robustness of model fitting. This was facilitated by leveraging Theano (The Theano Development Team et al., 2016), a numerical computation library widely used in deep learning at the time, whose automatic differentiation capabilities enabled these gradient-based algorithms.

The announcement of the discontinuation of Theano in 2017 prompted the PyMC project to adopt a new computational backend strategy. After considering other deep learning libraries, most notably TensorFlow and TensorFlow probability, the Theano project was forked into the non-affiliated Aesara project (Bastien et al., 2022a), and later the PyMC team forked Aesara once again into PyTensor (Bastien et al., 2022b). The focus of PyTensor is no longer to support deep learning, but instead to build, optimize, and compile symbolic computational graphs to serve the needs of PyMC.

PyMC’s explicit use of a computational graph also serves to distinguish it from systems like Pyro which implement inference using effect handlers and Stan where the program is compiled into a log density function in C++ with relatively less introspection of the program’s AST representation before doing so.

PyMC has played a significant role as an incubator for other libraries, nurturing and facilitating the development of specialized tools and functionalities. For example, ArviZ (Kumar et al., 2019) has emerged as a dedicated package for exploratory analysis of Bayesian models, providing a library-agnostic solution sampling diagnostics, model criticism, model comparison, and result preparation. Additionally, the handling of generalized linear models (GLMs) has found a new home in the Bambi library (Capretto et al., 2022), which has been able to focus on refining and expanding this functionality. For instance, Bambi also supports models such as splines, Gaussian processes, and distributional models. PyMC’s collaborative ecosystem has thus enabled the evolution and diversification of tools, allowing each library to excel in its respective domain while collectively advancing the field of Bayesian modeling.

The remainder of this article will focus primarily on the practical application of PyMC to building and fitting probabilistic models. We start with an API overview and then demonstrate the usage of PyMC through examples. We invited interested readers to visit https://www.pymc.io to learn more about PyMC through its extensive documentation and additional resources listed there. For those interested in implementation details we suggest this video https://www.youtube.com/watch?v=0B3xbrGHPx0.

We put special emphasis on how features of PyMC can help perform a modern Bayesian workflow. Section High-level architecture of PyMC briefly discusses PyMC’s main modules and the features they provide, and Section PyTensor discusses the most relevant aspects of PyTensor for PyMC users. Finally, we conclude with Section Discussion discussing the future of the PyMC package.

The version of PyMC used for this article is 5.5.0 All analyses are supported by extensive documentation in the form of interactive Jupyter notebooks (Kluyver et al., 2016) available in a GitHub repository (https://github.com/pymc-devs/paper_v5), enabling readers to re-run, modify, and otherwise experiment with the models described here on their own machines. This repository also includes instructions on how to set up an environment with all the dependencies required to replicate the contents of this article.

Api overview

Random variables (RVs)—variables whose values are described by a probability distribution—serve as the building blocks of a PPL. In PyMC, probability distributions are PyTensor RandomVariable instances that are created from subclasses of the Distribution class. These subclasses define the specific type of distribution, such as a normal (Normal) distribution or a binomial distribution (Binomial), and specify the parameters of the distribution as arguments, such as the mean and standard deviation for a normal distribution. The suite of probability distributions is organized into seven groups: continuous, discrete, multivariate, mixture, time series, censored1 , simulator, and symbolic distributions.

In general, users work with PyMC random variables as part of a larger model that describes how variables are related to each other and to the data. To this end, PyMC’s Model class is a container for all of the variables and other attributes that define a probabilistic model. The Model is a context manager that gathers the collection of interrelated observed and unobserved random variables as they are specified by the user. In this sense, an RV can be understood as the symbolic expression describing how a variable in the model is the result of mathematical operations on probability distributions. For example, an RV might be the distribution Normal(0, 1), or an expression such as the sum Normal(0, 1) + Uniform(2, 3).

Each random variable in a model can be either observed or unobserved, depending on whether it is associated with available data or is latent. An unobserved RV can be specified in PyMC by a name (string) and zero or more arguments, corresponding to the parameters of the statistical distribution. For example, a normal prior can be defined in a Model context like this:

 with pm.Model():

   x = pm.Normal(″x″, mu=0, sigma=1)

Here, the name “x” is the same as the assigned Python variable, but it need not be.

Observed RVs are defined similarly, but with an additional observed keyword argument to which the data is passed:

 with pm.Model():

   obs = pm.Normal(″x″, mu=0, sigma=1, observed=[-1, 0, 0, 1])

In this example, the list passed to the observed argument represents four samples of a one-dimensional random variable. The observed argument supports lists, NumPy arrays, Pandas Series, DataFrames, and PyTensor arrays. When used in combination with variational inference, observed supports minibatching using pm.Minibatch.

To describe a parent-child relationship within a model, random variables can be used as parameters of other random variables. For example, if we wish to assign a normal prior and an exponential prior to the mean and standard deviation, respectively, of a normal distribution:

 with pm.Model():

   mu = pm.Normal(″mu″, mu=0, sigma=1)

   sigma = pm.Exponential(″sigma″, 1)

   x = pm.Normal(″x″, mu=mu, sigma=sigma)

Random variables can be transformed and combined with other random variables algebraically using built-in operators and a suite of functions to yield deterministic nodes in the model:

 with pm.Model():

   x = pm.Normal(″x″, mu=0, sigma=1)

   y = pm.Gamma(″y″, alpha=1, beta=1)

   plus_2 = x + 2

   summed = x + y

   squared = x∗∗2

   sined = pm.math.sin(x)

Though these transformations work seamlessly, the resulting values are by default treated as intermediate and are not stored when model fitting. To store them, the pm.Deterministic wrapper should be applied, which gives the deterministic node a name and a value.

 with pm.Model():

   x = pm.Normal(″x″, mu=0, sigma=1)

   plus_2 = pm.Deterministic(″x plus 2″, x + 2)

Notice the name string passed to any variable need not be the same as the assignment target.

Models often require batches of RVs that have the same prior probability distribution. Users may be tempted to copy and paste an RV, or use a for iterator, but this leads to large and inefficient computation graphs:

 with pm.Model():

   # not recommended:

   x = [pm.Normal(f″x_{i}″, mu=0, sigma=1) for i in range(3)]

We instead recommend building a vector-valued RV, e.g.,:

 with pm.Model() as model:

   x = pm.Normal(″x″, mu=0, sigma=1, shape=3)  # length-3 array

x is now an array of length 3, representing three independent univariate normal variables2 . Additionally, it is possible to give different parameters to each of these Normal RVs by passing arrays of the appropriate size to the arguments mu and sigma. Usually, dimensions in probabilistic programs relate to real-world variables. So instead, an even better approach is to use the coordinates feature which allows the naming of dimensions:

 coords = {″city″: [″Santiago″, ″Mumbai″, ″Tokyo″]}

 with pm.Model(coords=coords) as model:

   # recommended:

   x = pm.Normal(″x″, mu=0, sigma=1, dims=″city″)

Here, we parameterize x using dims instead of shape. The Model associates the first dimension of x with the dimension named “city” in the coords argument, thereby creating the RV again as length 3 array. This information will also be associated with the output from model fitting, which simplifies working with the results.

Operations, including indexing, can be applied to vector-valued RVs in the same way one would operate on a NumPy array. Selection based on the dims and coords is not yet implemented.

 with model:

   y = x[0] * x[1]  # indexing is supported

   x.dot(x.T)  # linear algebra is supported

   x.sel(city=″Mumbai″)  # not yet supported

For each RV, PyMC automatically provides a sensible distribution-specific default initial value when starting the MCMC. These are usually means or medians, but can be customized via the initval parameter. This is sometimes helpful, for example when trying to identify problems with a model specification or initialization.

 with pm.Model() as model:

   x = pm.Normal(″x″, mu=0, sigma=1)

   y = pm.Normal(″y″, mu=0, sigma=1, initval=-3)

 # To get initial values for all RVs:

 model.initial_point()

 {′x′: array(0.), ′y′: array(-3.)}

A PyMC Model has references to all random variables (RVs) defined within it and provides access to the model log-probability and its gradients. Consider the following model:

 rng = np.random.default_rng (seed=0)

 x = np.linspace(-1, 1)

 y = rng.normal(x∗3, 1)

 with pm.Model() as model:

   a = pm.Normal(″a″, mu=0, sigma=1)

   b = pm.HalfNormal(″b″, sigma=1)

   mu = pm.Deterministic(″mu″, a + b∗x)

   obs = pm.Normal(″obs″, mu=mu, sigma=1, observed=y)

The Model retains collections of different subsets of variables as properties. To get all RVs, except for the deterministic ones:

 model.basic_RVs

 [a ~ Normal(0, 1), b ~ HalfNormal(0, 1), obs ~ Normal(mu, 1)]

To get all unobserved RVs, except for the deterministic ones:

 model.free_RVs

 [a ~ Normal(0, 1), b ~ HalfNormal(0, 1)]

To get all observed RVs:

 model.observed_RVs

 [obs ~ Normal(mu, 1)]

The joint log probability of the unobserved and observed variables is the sum of the log probability of each variable, taking dependencies into account:

logp(a,b,obs)= logp(a)+logp(b)+logp(obs|a,b)== −a22−log(2)+log(π)2logp   for   a ~ Normal(0, 1)−b22+log(2)−log(π)2logp   for   b ~ HalfNormal(1)+∑i(−(yi−(a+bxi))22−log(2)+log(π)2)logp  for  obsi ~ Normal(mu, 1) 

PyMC automatically transforms bounded variables into unbounded ones to facilitate efficient sampling. Transformations are registered to classes from which distributions can inherit. For example, the HalfNormal distribution inherits from the PositiveContinuous class which has a logarithm transformation registered to it. Here, a logarithm transformation can be applied to the half-normal b with R+ support, creating the variable b_log__=log(b) which has support in all of R. The transformed log probability is therefore:

logp(a,b_log_____,obs)=logp(a,b,obs)+log(dexp(b_log_____)db_log_____)=logp(a,b,obs)+b_log_____

This log probability computation and transformation are performed automatically by PyMC and accessible via the model method compile_logp:

 logp = model.compile_logp()

 logp({″a″: 0.5, ″b_log___″: 1.2})

 array(-73.39055683)

The compile_logp function optimizes and compiles the symbolic PyTensor computation graph into the selected backend (C, Numba or JAX), conditioned on the observed_RVs. The returned compiled function can be evaluated numerically for arbitrary values of the free RVs. The optimization and compilation of the computational graph done by the compile_logp method is expensive, but once this is done, the resulting compiled function can be called efficiently on different inputs of similar types.

Having fully specified a probabilistic model, we can select an appropriate method for inference. This may be a simple optimization, where we find the maximum a posteriori estimates for all model parameters:

 with model:

   fit = pm.find_MAP()

 fit[′a′], fit[′b′]

 (array(0.12646729), array(3.38501407))

Alternatively, we may choose a fully Bayesian approach and use Markov chain Monte Carlo to fit the model. This returns a trace of sampled values as output, representing the marginal posterior distribution of all model parameters. The sample function provides the interface to all of PyMC’s MCMC methods. For example, if we want 1,000 posterior samples after tuning the sampler for 1,000 iterations, we can call:

 with model:

   idata = pm.sample(1000, tune=1000)

 Auto-assigning NUTS sampler…

 Initializing NUTS using jitter+adapt_diag…

 Initializing NUTS using jitter+adapt_diag…

 Multiprocess sampling (4 chains in 4 jobs)

 NUTS: [a, b]

 Sampling 4 chains for 1_000 tune and 1_000 draw iterations

 (4_000 + 4_000 draws total) took 3 seconds.

idata, returned by pm.sample is an ArviZ InferenceData data object. This is a specialized data format based on a flexible N-dimensional array package, xarray (Hoyer & Hamman, 2017). The InferenceData object is designed to handle MCMC output3 : it stores the sampled values, along with other useful, diagnostic output from the sampling run. This facilitates the use of ArviZ for summarizing the estimated model, both in tabular and graphical form (see Fig. 1 for an example of a trace plot).

 az.plot_trace(idata, var_names=[′a′, ′b′])

 az.summary(idata, var_names=[′a′, ′b′])

Figure 1 Trace plot for parameters (A and B).

On the left, are kernel density estimates (KDE). On the right are the values of these parameters at each iteration.

High-level architecture of pymc

PyMC includes user-facing and internal features and APIs to support Bayesian modeling, including a wide variety of distributions and a suite of inference algorithms such as MCMC, sequential Monte Carlo (SMC) and Variational Inference (VI).

At the highest level, PyMC is organized into a set of quasi-independent but interoperating modules. This includes core components like models and probability distributions, inference algorithms such as MCMC step methods and variational approximations, and advanced modeling components like Gaussian processes and ordinary differential equations (Fig. 2). The architecture also depends on two other libraries, PyTensor (see section PyTensor) and ArviZ (Kumar et al., 2019), for computational backend functionality and exploratory analysis of Bayesian models, respectively.

Figure 2 PyMC architecture diagram highlighting the major modules.

Mean	SD	HDI_3%	HDI_97%	MCSE_mean	MCSE_SD	ESS_bulk	ESS_tail	R_hat	
a	0.129	0.139	−0.118	0.404	0.002	0.002	3,816.0	3,118.0	1.0	
b	3.385	0.237	2.938	3.815	0.004	0.003	4,211.0	2,654.0	1.0	

Distributions

Because Bayesian statistics involves constructing probabilistic models, PyMC provides robust implementations of all commonly-used probability distributions as classes in its distributions module, as well as a large set of more specialized distributions. The availability of these classes facilitates the construction of probabilistic graphs for forward sampling and inference. Stochastic random variables in PyMC models have a common set of properties as defined by the Distribution base class; these include methods for calculating log-probabilities, log-cumulative probabilities, and sampling random values. Distributions are broadly grouped according to whether they are continuous or discrete and univariate or multivariate, which confers additional properties to them according to this taxonomy (i.e., shape and data type values). In addition to these user-facing features, PyMC’s inference algorithms can manipulate distribution objects to improve their performance; for example, distributions that are bounded in their support are automatically transformed into real-valued spaces in order to improve sampling efficiency. For modeling time series, there are more complex multivariate distributions that account for serial dependence among elements of a sequence of variables, notably the autoregressive process and random walk processes (both Gaussian and Student-T). Also, there are wrapper classes that enforce truncation and censoring bounds for arbitrary base distributions which can be useful, for instance, in the construction of survival models. If a particular distribution of interest is not included in the distributions module, PyMC allows users to specify their own distributions.

Model

As introduced in Section API overview, the Model module provides the Model class, which encapsulates all components of the Bayesian model the user specifies. The user API for a Model instance is through a context manager, which serves to automatically add variables to the model graph, according to the associations between variables by the user; specifically, a random variable passed as an argument to another random variable establishes a parent-child relationship between them. All interaction with a Model instance by the user (e.g., model modification, model fitting) is done through the context manager. While the Model class has a large suite of methods, almost none of them are user-facing but are instead called by other methods and functions in PyMC in order to set up, introspect, or manipulate the model.

As a convenience to users, the Model module includes functions for generating visualizations of the model DAG (see for example Fig. 3), using either GraphViz (Gansner & North, 2000) or NetworkX (Hagberg, Schult & Swart, 2008).

Figure 3 Figure generated with the command pm.model_to_graphviz(mining).

Each random variable is represented by an ellipse which shows the name of the variable and its type. The rectangular boxes correspond to Deterministic variables, which are deterministic once their inputs are given (they do not add any randomness themselves). The numbers in the bottom-right corner of the rounded rectangles indicate the shape of the enclosed variable(s) (the number of independent variables that exist under this name): we see that the 111 years considered are split into two missing years and 109 observed. Finally, observed variables are colored in gray.

Logprob

This module contains the logic for operating with RandomVariable objects including: Converting RandomVariable graphs into joint log-probability graphs, transforming constrained RandomVariables so their support is on unconstrained spaces, RandomVariable-aware pretty printing, and LaTeX output.

Step methods

PyMC constructs posterior samplers using a Metropolis-within-Gibbs scheme, where blocks of parameters are assigned the same MCMC step algorithm. Distributions define their own default step algorithm, but this may be manually overridden by the user. For instance, in the coal mining disaster example (see Section Coal mining disasters) the Metropolis step method was assigned to the discrete sp variable and NUTS was assigned to the rest of the variables, which are continuous.

The default sampler for all continuous distributions is based on the No-U-Turn sampler (Hoffman & Gelman, 2014)4 . In practice, many models use only continuous parameters, and so will be only using this algorithm, which sees particular attention for performance. Some changes from the algorithm as originally described include multinomial sampling of the tree (Betancourt, 2016), and a corrected U-turn check. Other available inference methods include random-walk Metropolis (including specialized versions for binary and categorical parameters), a slice sampler (Neal, 2003), and Differential Evolution Metropolis sampling (Ter Braak, 2006).

Sampling

In addition to the MCMC step methods described in the previous section, which are useful to sample from the posterior distribution, PyMC supports sampling from both the prior predictive and posterior predictive distributions with flexible shapes allowing for in-sample estimates, as well as sample interpolation or extrapolation. This can be done using the pm.sample_prior_predictive or pm.sample_posterior_predictive functions. It is important to remark that PyMC allows using the same model definition to compute posteriors distributions (backward sampling) or predictive distributions (forward sampling), without requiring any intervention from the user.

Additionally, PyMC supports samplers from BlackJAX and NumPyro, as shown in Section JAX-based sampling.

Variational

The variational inference implementation is inspired by Ranganath et al. (2016) and defined as follows:

(1) supf∈ℱt(Eq[(Op,qf)(z)])

PyMC implements O—Operator, p,q—approximations, f—test function t—distance function family in a class structure; this is a nice abstraction over variational inference algorithms. One can construct standard Kullback-Leibler (KL) divergence with mean field approximation using this framework. To show that, let’s set operator OKLp,q such that

(2) OKLp,q:f↦(z↦logq(z)−logp(z|x)), ∀f∈ ℱ,

In a sense OKLp,q is constant over all f∈= ℱ, and yields a constant function without dependency on f. Using a short notation we can write

(3) (OKLp,qf)(z)=log⁡q(z)−log⁡p(z|x).

The simplified objective now does not need supf∈ℱ anymore, and setting the distance function to identity tI(x)=x we obtain

(4) supf∈ℱ tI=(E[(OKLp,q f)(z)])=Eqlogq(z)−logp(z|x)

Setting variational family q to the Mean Field family

(5) qMF(z)=N(z|μ,σ2)

completes the derivation.

Under this framework, PyMC also has Stein Variational Gradient (Liu & Wang, 2016) and β−KL (Burgess et al., 2018) objectives. Further extensions are made simpler for researchers and are open for contributions.

ODE

PyMC includes an Ordinary Differential Equations (ODE) module. The API mimics requires a function f(y,t,θ) which takes as arguments an array of states y, a time argument t, and an array of parameters θ. Once a solution to the ODE is found, an array of times at which to evaluate the solution is used.

It is implemented as an PyTensor Op which allows Hamiltonian Monte Carlo to differentiate through the ODE solution. The underlying implementation uses Scipy’s ODE solver which in turn uses the lsoda routine in the ODEPACK library written in FORTRAN.

SMC

Sequential Monte Carlo is a family of Monte Carlo methods. It has wide applications to Bayesian inference for static models and dynamic models, such as sequential time series inference and signal processing (Del Moral, Doucet & Jasra, 2006; Ching & Chen, 2007; Naesseth, Lindsten & Schön, 2019; Chopin & Papaspiliopoulos, 2020). PyMC supports sampling from static models using a Sequential Monte Carlo method, with many kernels, including, Metropolis-Hastings, Independent Metropolis-Hastings, Hamiltonian Monte Carlo. The SMC sampler can be useful for sampling multimodal posteriors, specially when modes are separated by very low probability regions. Additionally, the Independent Metropolis-Hastings kernel can be useful for models where the gradients are not available and for which complexity and/or dimensionality makes Metropolis-Hastings a poor choice. Moreover, SMC computes the marginal likelihood of a model as a by-product of sampling. In this module, we also find the code necessary to run Sequential Monte Carlo for Approximate Bayesian Computation. That is, models where the likelihood is not available explicitly, but we have a method to simulate synthetic data given a set of unknown parameters (Sisson, Fan & Tanaka, 2007; Sunn Sunnåker et al., 2013; Martin, Kumar & Lao, 2021).

Gaussian process

A Gaussian process (GP) can be used as a prior probability distribution over the space of continuous functions for modeling non-linear processes non-parametrically. PyMC includes a specialized API to define and fit GP models, specifically using latent and marginal approximations, and generate predictions for arbitrary inputs. The GP estimation is fully compatible with the MCMC, ADVI, and MAP methods, allowing users to flexibly fit models of various computational complexities while balancing computational estimation time.

Additionally, PyMC includes a number of kernels, including exponentiated quadratic, Matern, and periodic kernels, while allowing for the flexibility of users to define their own kernels if need be.

Case studies

Coal mining disasters

Between 1851 and 1962, a record number of accidents occurred in coal mines located in the United Kingdom (Jarrett, 1979). It is suspected that the application of certain safety regulations had the effect of reducing the number of accidents. Therefore, we are interested in estimating three quantities: the year in which the rate changed (the switch-point), the rate of accidents prior to regulation, and the rate after the regulation change.

The data is shown in Code Block 1, we have the variable disasters that contains the number of accidents per year and the variable years containing the range of years for which we have data. To encode the data, we are using a Pandas Series with np.nan values for the missing data.

 years = np.arange(1851, 1962)

 disaster_data = pd.Series([4, 5, 4, 0, 1, 4, 3, 4, 0, 6, 3,

                 3, 4, 0, 2, 6,

                3, 3, 5, 4, 5, 3, 1, 4, 4, 1, 5,

                 5, 3, 4, 2, 5,

                2, 2, 3, 4, 2, 1, 3, np.nan, 2,

                 1, 1, 1, 1, 3,

                0, 0, 1, 0, 1, 1, 0, 0, 3, 1, 0,

                 3, 2, 2, 0, 1,

                1, 1, 0, 1, 0, 1, 0, 0, 0, 2, 1,

                 0, 0, 0, 1, 1,

                 0, 2, 3, 3, 1, np.nan, 2, 1, 1,

                 1, 1, 2, 4, 2,

                0, 0, 1, 4, 0, 0, 0, 1, 0, 0, 0,

                 0, 0, 1, 0, 0,

                 1, 0, 1])

Code 1. Data for the coal mining disaster example.

How can we build a model for this problem? One approach is to think generatively, that is, create a story of how data may have been generated. Generative stories can be very powerful, informal devices that aid model construction and understanding (Blitzstein & Hwang, 2019).

We can think of our problem as having a moving slider, years to the left of this slider are assigned an average number of accidents while years to the right are assigned a different average number of accidents. A key property of Bayesian modeling is that we consider multiple plausible scenarios that could have generated the data (see Fig. 4).

Figure 4 Prior predictive check for the coal mining model.

The top panels (A) show one sample from the prior predictive distribution, including a visualization of the switching rate, which in this case is increasing (orange line). The bottom panels (B) show 500 samples from the prior predictive distribution. These prior predictions are within the realm of possibilities for this model, ignoring the data: this testifies to well-chosen priors. On the bottom left panel, we can see two vertical white lines (gaps), this corresponds to the missing observations.

Next, we need to specify prior probability distributions which quantify the information we have about plausible parameter values before we have observed any data. For the “slider” we will use a discrete uniform distribution that assigns equal probability to all years in a given interval, although other choices are possible. This distribution has two parameters: the lowest possible value, and the highest one. In our problem, those correspond to the year 1851 and 1962, respectively. A range wider than this does not make sense given that we only have data for this particular range, and a narrower range will imply that we have some external information indicating that not all years between 1851 and 1962 are equally plausible candidates.

Our observed data consists of counts, i.e., the number of disasters. Commonly, the Poisson distribution is used for count data. The Poisson distribution is defined using a single parameter that represents the average rate of events (disasters) in our example. As we do not know the rate and want to estimate it, we have to set a prior distribution for it. We do not have much information other than the rate must be positive and “most likely” have a small value, given that coal mining disasters are not very frequent events. One option could be to pick the distribution Expon(1), which says that the expected rate is 1 but is wide enough to allow for lower and higher values.

Using standard statistical notation, we can write the model as:

(6) sp~U(A0,A1)t0~Expon(1)t1~Expon(1)ratet={t0,if t<sp,t1,if t≥spacct Poisson(ratet)

And using PyMC we can write this model as described in Code Block 2.

 with pm.Model() as mining:

   # Prior for the switch-point

   sp = pm.DiscreteUniform(′sp′,

                 lower=years.min(),

                 upper=years.max())

   # Priors for the rate before (t_0) and after (t_1)

   t_0 = pm.Exponential(′t_0′, 1)

   t_1 = pm.Exponential(′t_1′, 1)

   # We assign the rates according to sp

   rate = pm.Deterministic(″rate″,

                 pm.math.switch(sp < years, t_0, t_1))

   # Likelihood

   acc = pm.Poisson(′acc′, rate, observed=disasters)

   # Backward and forward sampling

   idata = pm.sample()

   pm.compute_log_likelihood(idata)

   idata.extend(pm.sample_prior_predictive())

   idata.extend(pm.sample_posterior_predictive(idata))

Code 2. PyMC model for the coal mining disaster example.

Once defined in PyMC we can get a visual representation of the model as shown in Fig. 3. We can use this visual representation to check that we do not have a semantic error in the model and to communicate the model to others.

One remarkable feature of PyMC is that its syntax is very close to the statistical notation, as we can see by comparing Eq. (6) with Code Block 2. The cases in Eq. (6) are coded using the pm.math.switch (condition, true, false) function, which uses the first argument to select either of the next two arguments.

Missing values are handled concisely by passing a numpy.ndarray, pandas.DataFrame or pandas.Series with NaN values (see Code Block 1) to the observed argument when creating an observed stochastic random variable. This means that we will automatically get a posterior predictive distribution over the missing values. The imputed and observed values are combined into a Deterministic node acc that represents the original vector specified as an observed random variable.

Prior predictive checks

Specification of the prior distribution is of central importance in Bayesian modeling, but it is often difficult even for statistical experts (Mikkola et al., 2023). One problem when choosing priors is that it may be difficult to understand their effect as they propagate down the model into the data. The choices made in the parameter space may induce something unexpected in the observable data space (Martin, Kumar & Lao, 2021). Thus, a very helpful practice is to obtain samples from the prior predictive distribution. These are samples from the likelihood(s) of the model, but without conditioning on the observed data. Or, in simpler terms, predictions from the model before seeing the observed data. PyMC users can do this by calling the pm.sample_prior_predictive function. It is important to note that users do not need to write any extra code, or change the model as PyMC is capable of using the same model definition to compute posteriors distributions (backward sampling) and predictive distributions (forward sampling).

Code Block 3 shows one potential way of performing a prior predictive check. See Fig. 4 for the output of this code.

 ax[0,0].plot(years[np.isfinite(disasters)],

     idata.prior_predictive[″acc_observed″].sel(draw=50,

                           chain=0),

                           ″.″)

 az.plot_dist(idata.prior_predictive[″acc_observed″],

        ax=ax[1,1],

        rotated=True)

Code 3. Part of the code to generate Fig. 4. Lines related to the style of the plot have been omitted.

On the top panel, we can see one sample from the prior predictive distribution, for this sample the mean rate before the year 1880 is around 0.3, and after that around 1.3 (orange line). From the bottom panel, we can see that the average prior predictive prediction describes a uniform distribution of accidents across years, this is expected given that we have defined the same prior for t_0 and t_1. Additionally, just by eyeballing, we can see that our model favors relatively low values of accidents per year, with around 85 percent of the mass being assigned to values equal to or lower than 3. A more accurate estimate can be obtained by counting samples satisfying this property: np.mean(idata.prior_predictive[″acc_observed″] < 3).

Once confident with the model specification, we can estimate the parameters using one of the multiple inferential methods available in PyMC. If we decide to use Markov Chain Monte Carlo methods (MCMC), the continuous variables are sampled using an Adaptive Dynamic Hamiltonian Monte Carlo called NUTS (Hoffman & Gelman, 2014). Solving models with a discrete or a mix of discrete and continuous variables, like the one in Code Block 2 is also possible using compound samplers that could be manually specified by the user or automatically assigned by PyMC. For example, the variable sp in Model 2, being discrete, will be assigned to the Metropolis sampler, and the rate variables, continuous, to NUTS. Other samplers, such as Sequential Monte Carlo (SMC), are also suitable for a mix of discrete and continuous variables.

Posterior sampling

A common way to visually inspect the posterior is by plotting the marginal distributions for each parameter, as in Fig. 5. Once we have computed the posterior, we can use it to answer questions of interest: this can be done by computing numerical quantities, generating plots, and more often than not by a combination of both.

Figure 5 Trace plot generated with the command az.plot_trace(idata, combined=True), by default this command will read the posterior group from the idata object and generate, on the right a traceplot and, on the left, a histogram for discrete variables or a kernel density estimate (KDE), for continuous variables.

The combined argument is a flag for combining multiple chains into single histogram or KDE. If False (default), chains will be plotted separately.

From Fig. 6 we can see that the switch-point (orange line) is most likely around 1890, but we still have some uncertainty. The orange band represents the 94% credible interval and goes from 1885 to 1894. That is, according to the data and model, we think there is a 94% chance that the rate of accidents changed between 1885 and 1894. The black line represents the posterior mean for the rate of accidents, and the gray band is the 94% credible interval for that mean. Notice that in our model we specify prior distributions for two rate values, but we do not get just two point estimates for those rates, we get two distributions, one with mean ≈3 and one with mean ≈1. Even more, from approximately 1885 to 1894 we get a mix of those two distributions. The uncertainty about when the transition occurred is reflected in a rather smooth transition of the rates around these years.

Figure 6 The black line represents the mean rate of accidents for the different years, and the gray band represents the 94% HDI.

The orange line is the mean of the switchpoint and the band represents the 94% HDI.

Posterior predictive checks

In one of the previous sections, we saw we can sample from the prior and thus generate synthetic data representing the predictions from the model before seeing any data. In a similar fashion, we can also sample from the posterior in order to generate synthetic data conditioned on the observed data, i.e., predictions. These samples are known as posterior predictive samples and we can use them to check for auto-consistency. The generated data and the observed data should look like they were drawn from the same distribution, otherwise, we may have made a mistake during the model-building process or our model has some particular limitations. Posterior predictive checks can help us understand those limitations, and if necessary improve the model (Martin, 2018). There are potentially unlimited ways of performing posterior predictive checks. Figure 7 shows two different approaches using ArviZ.

Figure 7 Posterior predictive checks.

On the left panel, the observed data is in black, samples from the posterior predictive distribution are in blue, and their mean is in orange. This subplot was generated with az.plot_ppc(idata). On the right panel, the comparison is performed in terms of the LOO-PIT*. The ideal scenario is a uniform distribution, white line, for a finite dataset like the observed deviations within the grey band, are to be expected, and the blue line is the observed LOO-PIT. This subplot was generated with az.plot_loo_pit(idata, ″acc″). Both panels show that the model is well-calibrated. On the left, we can see that there is a good agreement between the observed and predicted data. On the right, we also see the agreement between observed and predicted data, but in such a way that perfect agreement will be a uniform distribution (white line). Due to the finite number of observations deviation from uniformity is expected. Because the blue line is inside the light blue band we can say that such deviations are to be expected. Note: *Leave-one-out probability integral transform.

Dirichlet-multinomial distribution

This example demonstrates the use of a Dirichlet compound multinomial distribution to model categorical count data. Models like this one are important in a variety of areas, including natural language processing (Madsen, Kauchak & Elkan, 2005), ecology (Harrison et al., 2019), and Genomics (Holmes, Harris & Quince, 2012; Nowicka & Robinson, 2016).

The Dirichlet-multinomial can be understood as taking draws from a multinomial distribution where each sample has a slightly different probability vector, which is itself drawn from a common Dirichlet distribution. In contrast with the multinomial distribution, which assumes that all observations arise from a single fixed probability vector. This enables the Dirichlet-multinomial to accommodate over-dispersed count data. Here we will discuss a community ecology example. Let’s assume we have observed counts of k=5 different tree species in n=10 different forests.

Our data is arranged in a two-dimensional matrix of integers where each row, indexed by i∈(0...n−1), is an observation (different forest), and each column j∈(0...k−1) is a category (tree species).

We could write this model as shown in Code Block 4, where we have a multinomial likelihood counts with a Dirichlet prior p. Furthermore, the prior is parameterized in terms of two hyper-priors, a Dirichlet distribution for the expected fraction of each category, frac and a log-normal conc controlling the concentration of the Dirichlet prior or in terms of the data the level of over-dispersion.

 with pm.Model() as model_dm_explicit:

   frac = pm.Dirichlet(″frac″, a=np.ones(k))

   conc = pm.LogNormal(″conc″, mu=1, sigma=1)

   p = pm.Dirichlet(″p″, a=frac ∗ conc)

   counts = pm.Multinomial(″counts″,

                 n=total_count, p=p,

                 observed=observed_counts))

Code 4. An explicit Dirichlet-multinomial distribution model.

The model in Code Block 4 is semantically and syntactically correct. Nevertheless, we can rewrite the model in an equivalent but slightly different form as shown in Code Block 5. The first change is that we are using pm.DirichletMultinomial distribution which takes into account that the Dirichlet distribution is conjugate to the multinomial and therefore there’s a computationally more efficient way to sample from, i.e. using a marginalized closed-form. The second modification with respect to Code Block 4 is that we are going to use labeled coordinates (coords) and dimensions (dims) which provides a tighter integration of PyMC with ArviZ.

 coords = {″tree″: trees, ″forest″: forests}

 with pm.Model(coords=coords) as model_dmm:

   frac = pm.Dirichlet(″frac″, a=np.ones(k), dims=″tree″)

   conc = pm.LogNormal(″conc″, mu=1, sigma=1)

   counts = pm.DirichletMultinomial(″counts″,

         n=total_count, a=frac ∗ conc,

         observed=observed_counts, dims=(″forest″, ″tree″))

   idata_dmm = pm.sample()

Code 5. A marginalized Dirichlet-multinomial model augmented with coordinates and dimensions.

Working with labeled arrays reduces the cognitive load of working with multidimensional arrays, reducing the chance of making errors and reducing frustration. In Code Block 5 by defining the frac Dirichlet parameters using dims="tree" we are guaranteed to have the dimensions of the prior matching the number of trees in our dataset. The advantage of using labels also extends to the post-inference processing stage. For example, idata_dmm.posterior.sel(tree="pine") will return the subset of the posterior related to pine and idata_dmm.posterior_predictive.counts.sel(tree="pine") will do the same for the predictive counts of pine.

Additionally, automatic labeling becomes possible. For instance, after sampling from the model in Code Block 5, calling az.plot_posterior(idata_dmm) generates Fig. 8. Notice how the frac parameter is meaningfully labeled with the names of the trees. The alternative would be integer labels with no intrinsic meaning.

Figure 8 Kernel density estimates of the predicted counts against the observed counts for each species.

JAX-based sampling

The most recent major version of PyMC is built on top of the PyTensor Python package, which allows the definition, optimization, and efficient evaluation of mathematical expressions involving multi-dimensional arrays. PyMC models, through PyTensor, can be compiled to C, Numba and JAX, and in principle, other computational backends could be added with relatively little effort. This allows for the efficient and fast evaluation of the log-probability density (see Section PyTensor for details). Still, the samplers accessible by calling pm.sample() are coded in Python and NumPy. A good approach to improve the performance of PyMC’s samplers is to write them in PyTensor. This will reduce the overhead of calling Python code and—more importantly—enable a series of optimizations due to PyTensor’s ability to manipulate graphs, including the ability to customize the sampler based on patterns in the model structure. Details of such optimizations are out of the scope of this article and will be discussed in a future manuscript.

An alternative to PyMC’s PyTensor-based samplers is samplers written in JAX. Using these samplers, all the operations needed to compute a posterior can be performed under JAX, reducing the Python overhead during sampling and leveraging all JAX performance improvements and features like the ability to sample on GPUs or TPUs. Currently, PyMC offers NUTS JAX samplers via NumPyro (Phan, Pradhan & Jankowiak, 2019) or BlackJAX (BlackJax devs, 2022) with the functions pm.sample_numpyro_nuts and pm.sample_blackjax_nuts, respectively. Significantly, BlackJAX and NumPyro can both be used because in PyMC the modeling language is decoupled from the inference methods; BlackJAX and NumPyro only require a log-probability density function written in JAX. This demonstrates that samplers can be developed independently of PyMC and then be made available to users of the library.

In the following example, we compare PyMC with its default Python/NumPy NUTS sampler, PyMC running the BlackJAX NUTS sampler, and PyMC running the NumPyro sampler. We also include cmdstanpy (Lee et al., 2017), a command line interface to Stan. The motivation for these comparisons is not to provide an exhaustive benchmark over a wide range of models and datasets but instead to give an example of the attainable speed-ups when using PyMC with a JAX-based sampler.

Suppose we are interested in ranking tennis players from 1968 until now5 . To do so, we can use the Bradley-Terry model. The central idea of this model is that each player has a latent skill θ. When players i and j play each other, player i wins with probability p(iwins∣θi,θj)=logistic(θi−θj). For example, if player i has a skill value of 1 and player j has a skill value of −1, then the Bradley-Terry model implies that the player i beats the player j with probability logistic(2)≈88.1%.

For the priors we set:

(7) σ∼N+(1)

(8) θi∼N(0,σ2)

when σ=1, 95% of the players’ skills distribution will fall between −2 and 2, which we consider quite plausible under this model and data, by setting a prior over σ we are also not ruling out smaller or larger distributions.

 with pm.Model() as tennis_model:

   sd = pm.HalfNormal(″sd″, sigma=1.0)

   skill_raw = pm.Normal(″skill_raw″, 0.0, sigma=1.0,

                shape=(n_players,)

   )

   skill = pm.Deterministic(″skill″, skill_raw * sd)

    logit_skill = skill[winner_ids] - skill[loser_ids]

   win = pm.Bernoulli(″win″, logit_p=logit_skill,

              observed=np.ones(winner_ids.shape[0])

   )

Code 6. A hierarchical, non-centered Bradley-Terry model for ranking tennis players.

We run the tennis_model in Code Block 6 using the NUTS sampler under six conditions: pymc: PyMC with the default sampler

cmdstanpy: CmdStanPy

pymc_numpyro_gpu_vectorized: PyMC with NumPyro NUTS sampler on the GPU, running chains using vmap.

pymc_numpyro_cpu_parallel: PyMC with NumPyro sampler on the CPU, running chains using pmap.

pymc_blackax_gpu_vectorized: PyMC with BlackJAX NUTS sampler on the GPU, running chains using vmap.

pymc_blackjax_cpu_parallel: PyMC with BlackJAX NUTS sampler on the CPU, running chains using pmap.

To see how the runtime changes with the size of the dataset, we choose different start years for the fits: 2020, 2019, 2015, 2010, 2000, 1990, 1980, and finally 1968. This means datasets ranging from 3,620 observations to 160,420.

For all conditions, we run 1,000 warm-up steps and 1,000 draws per chain, for a total of four chains.

Figure 9 shows the effective sample size per second for all the samplers previously mentioned. The values are an average of four separate runs. CmdStanPy performs better than PyMC on smaller datasets, but is slower on larger ones. PyMC with either the BlackJAX or NumPyro backends performs the best on the CPU, shown in yellow and magenta respectively. These JAX-based samplers have similar performance. On the other hand, when running on the GPU, the samplers are more efficient on larger datasets. Among the largest dataset, PyMC with NumPyro and BlackJax on the vectorized GPU performs the best, while PyMC with its default sampler and CmdStanPy (both on the CPU) show the worst results.

Figure 9 Effective sample size per second (ESS/s) for the tennis_model for different samplers (average over four independent runs).

Extensions

PyMC is a very flexible tool, and the PyMC community is quite active, the combination of which enables many specialized packages to be built by others. This provides the benefit of providing a coherent ecosystem of tools for PyMC users.

We note the following here: Bambi: BAyesian Model-Building Interface (BAMBI) in Python (Capretto et al., 2022) https://bambinos.github.io/bambi/.

beat: Bayesian Earthquake Analysis Tool (Vasyura-Bathke et al., 2020) https://github.com/hvasbath/beat.

calibr8: A toolbox for constructing detailed observation models to be used as likelihoods in PyMC (Helleckes et al., 2022; Osthege, Helleckes & Siska, 2022) https://calibr8.readthedocs.io.

CausalPy: A package focusing on causal inference in quasi-experimental settings https://github.com/pymc-labs/CausalPy.

Exoplanet: a toolkit for modeling of transit and/or radial velocity observations of exoplanets and other astronomical time series (Foreman-Mackey et al., 2021) https://github.com/exoplanet-dev/exoplanet.

pyei: Ecological inference, with an emphasis on racially polarized voting (Knudson, Schoenbach & Becker, 2021) https://github.com/mggg/ecological-inference.

pymc-bart: Bayesian additive regression trees (BART) for probabilistic programming (Quiroga et al., 2022) https://www.pymc.io/projects/bart.

SunODE: Fast ODE solver, much faster than the one that comes with PyMC (Seyboldt et al., 2022) https://github.com/pymc-devs/sunode.

Besides the main PyMC package, the PyMC developers also maintain a PyMC-experimental package, a collection of features extending the core functionality of PyMC in diverse directions. PyMC-experimental is intended to host unusual probability distributions, advanced model fitting algorithms, innovative yet not fully tested methods, or any code that may be inappropriate to include in the PyMC repository, but may want to be made available to users. This package is ideal for researchers and developers wanting to contribute new research as features to PyMC.

At the time of writing, PyMC-experimental includes A method to aggregate large datasets in order to speed up inference,

A method to approximate a posterior to a multivariate normal for use as a prior of a subsequent analysis

Three efficient reduced-rank methods for Gaussian processes: the Karhunen-Loeve expansion, deterministic training conditionals (DTC) (Quinonero-Candela & Rasmussen, 2005), and the Hilbert space GPs (Solin & Särkkä, 2020), and

The Pathfinder variational inference algorithm (Zhang et al., 2022).

Pytensor

PyTensor is a pure-Python library that allows one to define, optimize, and efficiently evaluate mathematical expressions involving multi-dimensional arrays, including automatic differentiation. It is used as the computational backend of the PyMC library and was developed from its predecessors Theano (The Theano Development Team et al., 2016) and Aesara (Bastien et al., 2022a). At its core, PyTensor implements an extensible computational graph framework that is accompanied by graph rewriting optimizations and linkers to various compilable backends. At the time of writing, PyTensor graphs can be readily linked to backends including C (Kernighan & Ritchie, 1988), JAX (Bradbury et al., 2018), and Numba (Lam, Pitrou & Seibert, 2015), yielding compiled functions that are much faster to evaluate than pure Python implementations of the computational graph. It combines aspects of a computer algebra system (CAS) with aspects of an optimizing compiler. This combination of CAS with optimizing compilation is particularly useful for tasks in which complicated mathematical expressions are evaluated repeatedly, and evaluation speed is critical, as is the case in MCMC applications. PyTensor does not only provide a powerful computation backend for PyMC, but also decouples PyMC from the underlying compilation backends, making it easier to use new compilers without disrupting the existing PyMC code-base.

Tensors

Now we will discuss some core concepts of PyTensor and how they relate to PyMC using a few simple code examples. For a more detailed description of PyTensor please refer to the official documentation (https://PyTensor.readthedocs.io/en/latest/).

To begin, in lines 1 and 2 of Code Block 7 we define two PyTensor tensors, the first one being a scalar (0 dimension) and the second being a vector (1 dimension). On line 3, a new tensor z is created by adding x and y, and then taking the natural logarithm. Then, on line 4 we add a name to the newly created tensor, as this can be useful to easily reference tensors when working with many of them.

It is important to note that contrary to other objects like Python integers, floats or NumPy arrays, the operations described in Code Block 7 do not immediately lead to numerical values. Instead, they are just a symbolic specification of the operations we want to compute. This example literally just defines the abstract expression, take the logarithm of the sum of x and y. To actually perform computations, we first need to define a function, and then call it with the proper input as shown in lines 6 and 7.

 x = pt.scalar(name=″x″)

 y = pt.vector(name=″y″)

 z = pt.log(x + y)

 z.name = ″log(x + y)″

 f = pytensor.function(inputs=[x, y],  outputs=z)

 f(x=0, y=[1,  np.e])

 >>> array([0., 1.])

Code 7. Definition and call of a PyTensor function. Notice that the tensors x, y, and z have been previously defined. When debugging it may be useful to avoid defining a function and instead perform a direct evaluation of the tensor, like z.eval(x: 0, y:[1, np.e]).

This separation of the abstract definitions of mathematical expressions and the actual computation of those expressions is central to PyTensor and hence PyMC. When defining a PyMC model, we are just defining a PyTensor computational graph that we will later use to obtain quantities like prior predictive samples, posterior samples, log-probabilities, etc. This separation is useful as PyTensor can automatically optimize the mathematical operations inside a graph. For example, if we define w = pt.exp(pt.log(x + y)), PyTensor will first simplify the graph to w = x + y and then perform the computation. Other optimizations include constant propagation, replacing numerically unstable operations with numerically stable versions, avoiding computing the same quantity more than once, and efficient sparse matrix multiplication.

Random variables

We now show how to manually generate samples from PyMC distribution and evaluate their log-probability. On line 1 of Code Block 8 we define a Normal distribution with mean 0 and standard deviation 1. On line 2 we take 1,000 draws from that distribution, using the pm.draw(.) function. Finally, on line 3 we use the pm.logp(.) function to compute the log-probability of the samples generated in the previous step. We use the eval() method to obtain the actual values, instead of a symbolic representation. Figure 10 shows the results.

Figure 10 Top panel kernel density estimate of the sample generated in Code Block 8, bottom panel scatter plot of the x values and their log probability.

 x = pm.Normal.dist(mu=0, sigma=1)

 x_draws = pm.draw(x, draws=1_000)

 x_logp = pm.logp(rv=x, value=[x_draws]).eval()

Code 8. In line 1 a PyMC distribution is used to specify a symbolic random variable from a corresponding to a Normal distribution. The pm.draw(.) call in line 2 invokes PyTensor graph compilation while accounting for automatic updating of random number generators, and returns an array with 1,000 draws of variable x. Using pm.logp(.), the log-probability densities of x for each element in x_draws are derived symbolically, and then evaluated by the call to the .eval() method.

While the example from Code Block 8 may seem trivial, it is important to note that the variable x which is passed to pm.draw(.) or pm.logp(.) can be the result of a symbolic computation involving multiple random variables and other tensors. A simple example is given in Code Block 9 where the random variable b depends on another random variable a, and the variable x is a tensor variable that merely depends on other variables, some of which represent random variables. To a much larger extent, this is how computational graphs of PyMC models are handled behind the scenes, and how users can access properties such as the log-probability densities of probabilistic graphs built with PyTensor.

 a = pm.Uniform.dist()

 b = pm.Normal.dist(mu=a, sigma=1)

 x = b + pt.as_tensor([2, 3, 4])

 x_draws = pm.draw(x, draws=1_000)

 x_logp = pm.logp(rv=x, value=[x_draws]).eval()

Code 9. Random variables such as the ones defined in lines 1 and 2 behave like tensor variables and can be used as such in standard operations such as addition (line 3). The sampling and derivation of log-probabilities (lines 4 and 5) can operate on any graph that involves random variables.

Discussion

PyMC has been the leading probabilistic programming language in Python for years. Its intuitive syntax that balances simplicity with flexibility has been key to this success. The contributor community is varied, composed of users, technical writers, developers, and even visual designers for artifacts such as logos. This diversity of contributors has aided in many ways to the improvement of the library and its adoption by a large audience. As PyMC has grown, its functionality has spun off into more specialized and feature-rich packages for the Bayesian community. For instance, sampling diagnostics, model comparison, and visualizations had been forked into ArviZ which supports PyMC but many other PPLs as well. Similarly, the definition of complex generalized linear hierarchical models using a formula notation similar to those found in R has now been delegated to Bambi.

In this way, the PyMC contributor environment has been incredibly beneficial for the computational Bayesian community. This is evidenced by the numerous sister packages that PyMC has seeded, each with a more focused developmental process. This makes it easier to maintain the software, add new features, and for the users to find specialized packages to fit their needs while continuing to grow an ever larger and interconnected community.

In this manuscript, we have highlighted some of the most relevant features of the current state of PyMC development and mention some changes to come in the near future. We trust that the technical innovations, strong community, and interoperability with the Scientific Python ecosystem herald a bright future for PyMC.

Contributing to pymc

As a community-driven project, we are always excited to collaborate with new contributors. For those interested in working on PyMC, we invite them to read our contributing guidelines (https://www.pymc.io/projects/docs/en/latest/contributing/index.html). As part of our quality control process, contributions are submitted as a pull request (PR) that is subject to review and revision prior to being merged into the appropriate project repository; most PRs need approval from at least 1 core developer. Major innovations and changes to the API are subject to collective agreement from the core contributors; see https://www.pymc.io/projects/docs/en/latest/contributing/ for how exactly we delineate responsibilities and decision-making power for each community role.

There are several ways to contribute to PyMC, and not all of them require familiarity with the code base. Notably, help with maintaining and improving documentation and the contribution of new case studies (https://github.com/pymc-devs/pymc-examples) are always welcome. Porting existing code from Bayesian books that are written using other PPLs is also encouraged (https://github.com/pymc-devs/pymc-resources). We also welcome talks and tutorials involving PyMC and related packages (https://pymcon.com/). We encourage anyone looking to get started to get in touch with the development team via Discourse (https://discourse.pymc.io/).

Software used in this article

Libraries explicitly used in the code examples: ArviZ (Kumar et al., 2019; Martin et al., 2022), Graphviz (Gansner & North, 2000), Jupyter (Kluyver et al., 2016), Matplotlib (Hunter, 2007; Caswell et al., 2022), NumPy (Harris et al., 2020), Pandas (McKinney, 2010; Pandas Development Team, 2022), PyTensor (Bastien et al., 2022b), SciPy (Virtanen et al., 2020; Gommers et al., 2022), xarray (Hoyer & Hamman, 2017; Hoyer et al., 2022). And PyMC itself, version 5.0.1 (Wiecki et al., 2022).

Appendix

PyMC is available from the Python Package Index at https://pypi.org/project/pymc/. Alternatively, it can be installed using conda, which is the recommended way of installing PyMC. The project is hosted and developed at https://github.com/pymc-devs/pymc. The package documentation, including installation instructions and many examples of how to use PyMC to conduct different statistical analysis, can be found at https://docs.pymc.io.

We thank Google Summer of Code (GSoC), a global program that offers student developers stipends to write code for open-source projects. We also want to thank all the students that participated in the GSoC and contributed to PyMC. We thank Martin Ingram who is the original author of the model in Section JAX-based sampling, the original model and benchmark can be found at https://martiningram.github.io/mcmc-comparison/ and Kevin Murphy for his helpful comments on an earlier version of this manuscript. We want to thank Adrian Seyboldt for his significant contributions to PyMC. Finally, PyMC would not be the same without the work of hundreds of volunteers reporting issues, fixing bugs, and contributing features to the project, to whom we are also indebted.

Additional Information and Declarations

Competing Interests

Author Contributions

Data Availability

1 This allows to transform regular PyMC distributions into censored ones.

2 Not to be confused with a multivariate normal of size 3, which would be created with pm.MvNormal.

3 For example on how to perform common operations you can read https://python.arviz.org/en/stable/getting_started/WorkingWithInferenceData.html.

4 This is the first description of the sampler, but the implementation in PyMC includes several improvements that have been developed since then, for example on mass matrix tuning.

5 This model and benchmarks were initially run by Martin Ingram and can be found on https://martiningram.github.io/mcmc-comparison/.

The authors declare that they have no competing interests.

Colin Carroll, Ravin Kumar and Junpeng Lao are employed by Google Inc., Christopher J. Fonnesbeck is employed by Baseball Operations Research and Development, Maxim Kochurov, Ricardo Vieira and Thomas Wiecki are employed by PyMC Labs and Michael Osthege is employed by Forschungszentrum Jülich GmbH.

Oriol Abril-Pla conceived and designed the experiments, performed the experiments, analyzed the data, performed the computation work, prepared figures and/or tables, authored or reviewed drafts of the article, and approved the final draft.

Virgile Andreani conceived and designed the experiments, performed the experiments, analyzed the data, performed the computation work, prepared figures and/or tables, authored or reviewed drafts of the article, and approved the final draft.

Colin Carroll conceived and designed the experiments, performed the experiments, analyzed the data, performed the computation work, prepared figures and/or tables, authored or reviewed drafts of the article, and approved the final draft.

Larry Dong conceived and designed the experiments, performed the experiments, analyzed the data, performed the computation work, prepared figures and/or tables, authored or reviewed drafts of the article, and approved the final draft.

Christopher J Fonnesbeck conceived and designed the experiments, performed the experiments, analyzed the data, performed the computation work, prepared figures and/or tables, authored or reviewed drafts of the article, and approved the final draft.

Maxim Kochurov conceived and designed the experiments, performed the experiments, analyzed the data, performed the computation work, prepared figures and/or tables, authored or reviewed drafts of the article, and approved the final draft.

Ravin Kumar conceived and designed the experiments, performed the experiments, analyzed the data, performed the computation work, prepared figures and/or tables, authored or reviewed drafts of the article, and approved the final draft.

Junpeng Lao conceived and designed the experiments, performed the experiments, analyzed the data, performed the computation work, prepared figures and/or tables, authored or reviewed drafts of the article, and approved the final draft.

Christian C Luhmann conceived and designed the experiments, performed the experiments, analyzed the data, performed the computation work, prepared figures and/or tables, authored or reviewed drafts of the article, and approved the final draft.

Osvaldo A Martin conceived and designed the experiments, performed the experiments, analyzed the data, performed the computation work, prepared figures and/or tables, authored or reviewed drafts of the article, and approved the final draft.

Michael Osthege conceived and designed the experiments, performed the experiments, analyzed the data, performed the computation work, prepared figures and/or tables, authored or reviewed drafts of the article, and approved the final draft.

Ricardo Vieira conceived and designed the experiments, performed the experiments, analyzed the data, performed the computation work, prepared figures and/or tables, authored or reviewed drafts of the article, and approved the final draft.

Thomas Wiecki conceived and designed the experiments, performed the experiments, analyzed the data, performed the computation work, prepared figures and/or tables, authored or reviewed drafts of the article, and approved the final draft.

Robert Zinkov conceived and designed the experiments, performed the experiments, analyzed the data, performed the computation work, prepared figures and/or tables, authored or reviewed drafts of the article, and approved the final draft.

The following information was supplied regarding data availability:

Code files are available at GitHub and Zenodo:

https://github.com/pymc-devs/paper_v5.

Osvaldo A Martin, Ravin Kumar, Oriol Abril-Pla, & Thomas Wiecki. (2023). pymc-devs/paper_v5: submission 1 (submission_1). Zenodo. https://doi.org/10.5281/zenodo.8121048.

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
