# Peer review of "PyMC: a modern, and comprehensive probabilistic programming framework in Python"

_PeerJ Computer Science, doi:10.7717/peerj-cs.1516_

## Round 0.1 · original submission · Minor Revisions

Based on the reviewers’ comments, you may resubmit the revised manuscript for further consideration. Please consider the reviewers’ comments carefully and submit a list of responses to the comments along with the revised manuscript.

Reviewer 1 ·

Basic reporting

The authors have presented PYMC, which is a Python library for Bayesian statistical modeling and probabilistic programming. Authors have provided a brief history of past versions and the necessities to update the library.

It is appreciated that new models are included in the distribution such as Time Series and GLMs are supported by Bambi however some community members wish to be part of PYMC. Authors should discuss briefly the selection process for inclusion of new models into the library.

Code block 3 is not referenced in text. Code blocks are not referred in order in the text. The code block 1 presents 115 values. The range suggests it should have 112 values.

Figure 1 and 9 is not referenced in the text. The presentation of Waveform in Figure 1 may be improved for clarity similar to Figure 5.

Edges of Figure 2 should be labeled to improve readability.

Figure 3-11 are not placed appropriately in the document ( far from the referenced text).

The caption of Figure 3 appears to be misleading. "The rectangular boxes with rounded corners indicate the shape of the variables.."

Figure 7 does not contain Y-axis label.
Caption of Figure 9 needs to be updated.
Figure 11 top panel may be decorated appropriately.

Experimental design

None

Validity of the findings

The title of the article suggest that the updated library has better performance however no evidence of performance is provided. It is suggested to include benchmark the performance in comparison with previous versions as well as current competitors.

Cite this review as

Reviewer 2 ·

Basic reporting

This article proposes a PyMC, which is a probabilistic programming library for Python that provides tools for constructing and fitting Bayesian models. It offers an intuitive, readable syntax that is close to the natural syntax statisticians. This seems to be a nice and wonderful contribution for the researchers to create and implement their models using this library e.g PyMC. One of the notable strengths of PyMC is its integration with PyTensor, a symbolic computation library, enabling compilation into various computational backends such as C, JAX, and Numba. This flexibility grants users access to different computational architectures, including CPU, GPU, and TPU.The authors has described the library in details along with sufficient information and examples. The article is very good and easy to read and understand, I am really impressed by the proposed framework. I have few suggestions if they can be incorporated to further strengthen the article.

1. The authors have discussed the case study of Coal mining disasters, I believe its discussed in details but it is recommended that that the authors may also discuss some other case studies to further validate the framework.
2. The authors rightly emphasize the intuitive and readable syntax of PyMC, which simplifies the modeling process and allows statisticians to describe models more naturally. This feature makes PyMC accessible to a wider audience and reduces the learning curve for those new to probabilistic programming.
3. While the article effectively covers the essential aspects of PyMC, it would be beneficial to include more specific details about its implementation, such as the availability of comprehensive documentation, community support, and integration with other Python libraries commonly used in data science and machine learning workflows.
4. Overall, I would rate this research article highly important, The comprehensive coverage of PyMC's features, coupled with its practical examples and emphasis on the open-source ecosystem, make it an excellent contribution to the field of probabilistic programming

Experimental design

See above

Validity of the findings

See above

Additional comments

see above

Cite this review as

---

## Round 0.2 · accepted · Accept

Congratulations, the revisions are satisfactory and the paper is recommended for publication.